# A Partial-Reconfiguration-Enabled HW/SW Co-Design Benchmark for LTE Applications

**Ali Hosseinghorban** * and **Akash Kumar** *

CFAED (Center for Advancing Electronics Dresden), Technische Universität Dresden, 01062 Dresden, Germany
* Correspondence: ali.hosseinghorban@mailbox.tu-dresden.de (A.H.); akash.kumar@tu-dresden.de (A.K.)

**Abstract:** Rapid and continuous evolution in telecommunication standards and applications has increased the demand for a platform with high parallelization capability, high flexibility, and low power consumption. FPGAs are known platforms that can provide all these requirements. However, the evaluation of approaches, architectures, and scheduling policies in this era requires a suitable and open-source benchmark suite that runs on FPGA. This paper harnesses high-level synthesis tools to implement high-performance, resource-efficient, and easy-maintenance kernels for FPGAs. We provide various implementations of each kernel of PHY-Bench and WiBench, which are the most well-known benchmark suites for telecommunication applications on FPGAs. We analyze the execution time and power consumption of different kernels on ARM processors and FPGA. We have made all sources and documentation public for the benefit of the research community. The codes are flexible, and all kernels can easily be regenerated for different sizes. The results show that the FPGA can increase the speed by up to 19.4 times. Furthermore, we show that the power consumption of the FPGA can be reduced by up to 45% by partially reconfiguring a kernel that fits the size of the input data instead of using a large kernel that supports all inputs. We also show that partial reconfiguration can improve the execution time for processing a sub-frame in the uplink application by 33% compared to an FPGA-based approach without partial reconfiguration.

**Keywords:** 5G; partial reconfiguration; benchmark

## 1. Introduction

Nowadays, wireless communication systems must support services such as virtual reality, 3D video communication, online games, IoT applications, autonomous vehicles, machine translation, and smart-grid automation. To this end, these systems must support high data rates, massive connectivity, low transmission delay, and high bandwidth. They must also adapt to frequent workload changes due to the high mobility of connected devices in the network [1,2]. FPGAs are well-known platforms for handling services with high throughput demands, because of their high parallelization capability [3]. Furthermore, partial reconfiguration (PR), also known as dynamic function exchange (DFX), improves the system's flexibility. With PR, the system can change part of the FPGA functionality while other parts are working [4,5]. Therefore, in the case of peak data rates, the system can increase the computational power by configuring a more parallel and faster module with high signal activity on the FPGA. In the case of low data rate requests, the system can configure a small, more sequential module with low signal activity on the FPGA to reduce power consumption. Therefore, FPGAs offer high parallelization and adaptivity for developing power-efficient and high-throughput services and applications [6–8].

Developers must evaluate kernels, mapping/scheduling policies, and application-level decisions to design efficient mobile services. Although the PHY-Bench [9] and WiBench [10] benchmark suites provide various LTE kernels, they were developed for general-purpose processors with high-level languages such as C/C++. However, developing and testing mobile communications services for FPGA is more challenging than for general-purpose

CPUs. Therefore, developing new standards and updating the latest versions of kernels developed using hardware description languages (HDLs) have higher costs and require more time. In this regard, we harnessed the Vivado High-Level Synthesis (HLS) [11] tool to convert the most famous LTE kernels from PHY-Bench and WiBench benchmark suites into HDL modules. Therefore, it was possible to modify, test, and add new features with only a meager cost [12]. This is because the developer can modify and test the kernels in C/C++ languages instead of HDL.

It is important to mention that, although HLS facilitates the procedure of converting code developed in C/C++ to an HDL module, it usually results in low-performance kernels because the codes are designed for sequential execution [13]. Therefore, we refactored the structure of the kernels to enable dataflow optimization. Dataflow optimization provides the opportunity for function-level pipelining and significantly improves the throughput and latency. Another important parameter that must be considered is the effect of partial reconfiguration on the system. Partial reconfiguration can improve the system's energy efficiency, especially when the system works under highly variant workloads. Therefore, we provided a suitable HDL wrapper for each kernel to enable partial reconfiguration in the system. With these wrappers, the system can replace the kernel that is currently working on the FPGA with another kernel. Finally, we monitored the real-time power consumption of the FPGA and analyzed the power–performance trade-off for different implementations.

Our main contributions in this paper are as follows:

- We developed an efficient HLS based-module for each kernel in PHY-Bench and WiBench benchmark suites using Vivado HLS. Furthermore, to improve the concurrency and parallelization in each kernel, we refactored the C/C++ implementations and changed them to apply dataflow optimization. We made the source code available, and researchers can easily modify kernels and regenerate all kernels with different sizes.
- We provided an HDL wrapper for each kernel with two AXI-Stream interfaces to receive input data and send output data through DMA. The wrapper also has an AXI-Lite interface to send and receive control and status signals. The wrappers for all kernels have the same interface ports. Therefore, we can swap all kernels in the FPGA during the run time with the help of partial reconfiguration.
- We compared each kernel's execution time and power consumption during execution on the ARM processor and on the ZynqMP SoC. To this end, we exploited the Ultra96-V2 by Avnet, which is an ARM-based, Xilinx Zynq UltraScale+ MPSoC development board [14], to run different kernels on an XCZU3EG-SBVA484 FPGA and an ARM Cortex-A53 processor. It is important to mention that both the ARM Cortex-A53 and XCZU3EG-SBVA484 FPGA are integrated on the same chip.

The rest of the paper is organized as follows. In Section 2, we briefly discuss some basic concepts in designing with ZynqMP SoCs. In Section 3, we review the related work. Section 4 discusses the characteristics of different kernels and how we synthesize them. In Section 5, we implement the kernels on a real platform and evaluate the system's power consumption and execution time for both hardware and software. Section 6 discusses the effect of partial reconfiguration, and we conclude the paper in Section 7.

## 2. Preliminaries

In this section we provide a brief explanation of the Zynq Ultrascale+ processing system, the AXI bus, partial reconfiguration, and the PCAP interface.

### 2.1. Zynq UltraScale+ MPSoC

The Zynq UltraScale+ MPSoC (ZynqMP) is a Xilinx product that integrates an FPGA, two to four ARM Cortex-A53s, and two Cortex-R5s on a single chip. ZynqMP SoCs are so powerful that it is possible to process hundreds of gigabits of data per second. These systems can be used for a variety of applications such as 5G, industrial IoT, etc. The ZynqMP has two parts. The first part, called the processing system (PS), contains ARM

processors. It is possible to execute C/C++ applications or even boot the Linux operating system on the PS part. The second part is the programmable logic (PL) part, which can be used to execute RTL modules.

### 2.2. Partial Reconfiguration

Partial reconfiguration or dynamic function exchange allows FPGA developers to design a system where the functionality of a part of the PL can be changed while the other parts of the PL are active. To this end, the Vivado design tool generates a partial bitstream for each reconfigurable module in addition to a full bitstream. Hence, at first, the PL is programmed with the full bitstream. Then, during the execution, the PS can partially reconfigure a module in the PL by programming the module's partial bitstream file. There are various ways to partially reconfigure the PL. In this paper, we used a process configuration access port (PCAP) method, because it is fast and does not require any additional logic in the PL.

### 2.3. Advanced Extensible Interface (AXI) Bus Interface

AXI is a high-performance bus interface, used for on-chip communications. In Xilinx Vivado, there are three types of AXI interfaces: AXI-MM (memory mapped), AXI-Lite, and AXI-Stream. AXI-MM is a simple bidirectional memory mapped data and address bus interface that has the capability for supporting burst reads and writes. AXI-Lite is a simple version of AXI-MM which does not supports burst reads and writes. It is usually used for sending control signals and receiving the status signal. AXI-Stream, on the other hand, is a fast, address-less unidirectional protocol used for transferring large quantities of data from master modules to slave modules.

## 3. Related Work

ASIC-based accelerators provide satisfactory performance and power consumption; however, they only execute a fixed program, and it is hard or impossible to change their functionality [15]. Considering new demands and rapid technology evolution in LTE and 5G applications, the use of ASIC entails considerable costs due to the lack of flexibility [16]. Venkataramani et al. [15] proposed SPECTRUM, a predictable many-core platform for LTE applications. The platform contains up to 256 lightweight ARM-based cores. Each core has a private scratchpad memory, and a software-controlled network on chip (NoC) connects all cores. As we show in this paper, due to the high parallelization capability of FPGAs, the execution time and power consumption of LTE applications on FPGAs is much lower than on ARM-based processors. Venkataramani et al. [17] also proposed a synchronous data flow (SDF) compiler toolchain to improve the utilization of the system by harnessing fine-grain scheduling.

Wittig et al. [18] pointed out the new performance demands and increasing parameter space in new generations of mobile networks. They showed that the use of FPGA and partial reconfiguration in communication applications could significantly improve the system's energy efficiency and reduce the sub-frame drop rate, due to the workload's adaptivity. Chamola et al. [19] surveyed various 5G applications implemented on FPGA. They discuss the effect of FPGA on the performance and the energy consumption of the system and how PR can improve them. Dhar et al. [20] proposed an integer linear programming (ILP)-based scheduling to map tasks for any application on FPGA using PR.

The most well-known benchmarks for communication applications are PHY-Bench [9] and WiBench [10]. These two benchmark suites provide various kernels commonly used in communication applications and standards such as WCDMA and LTE. These benchmarks are developed in C and C++ languages for general-purpose processors. Liang et al. [21] exploited HLS to convert some of the WiBench kernels into HDL modules. However, their modules and codes are not publicly available. In this paper, we also used HLS to convert all PHY-Bench and WiBench kernels to HDL modules, and we provide the source code to help the research community explore the effect of FPGA in communication applications.

## 4. Kernel Characteristics

This section analyzes the characteristics of different kernels of PHY-Bench and WiBench and discusses how the kernels are developed on the FPGA in detail. The Vivado High-Level Synthesis tool is a part of Xilinx Vivado Design Suite, which enables the developer to develop their modules using C, C++, or SystemC languages and transform them to RTL modules. The generated RTL modules can be directly implemented on Xilinx products. HLS improves productivity since the developer can design and test the modules faster in high-level languages than in RTL. Furthermore, the developer can rapidly explore different design alternatives with the help of some directives in HLS, in order to choose the best design. In this study, we modified the source code of LTE benchmarks in the C language and we used the Vivado High-Level Synthesis tool to compile them to RTL modules and test them.

The first step in making C/C++ code ready to be synthesized to an RTL module is to eliminate all the system calls such as "print to the console", "open a file", etc. In addition, all the dynamic memory allocations in the code should be replaced with static memory allocations. Although these changes make the code synthesizable, the generated RTL module has a very low performance. Therefore, HLS provides several primary directives such as pipelining, loop unrolling, or array partitioning to improve the module performance. These directives increase the parallelism in the code and reduce the latency of the generated HDL module. To be more specific, we discuss these three directives in more detail as follows:

- Loop unroll: this directive takes a variable called "Factor" which indicates how much the designer wants to unroll the loop. Assuming that Factor is set to N, then the HLS compiler creates N copies of the loop body. Therefore, the generated RTL module runs N iterations of the loop concurrently. Hence, the number of sequential iterations is reduced by factor of N.
- Pipeline: this directive divides the body of loop or function into a set of pipes (sections) and allows all sections to be run in a concurrent manner. This directive does not improve the execution time of a single iteration of a loop. However, it improves the input interval of the loop. This directive is very effective for loops where the dependency between operations is low and the number of iterations is high.
- Array partition: by default, the HLS compiler implements each array in the code with one large memory with one or two ports to access the data. The array partitioning divides the array into two or more smaller memories, which increases the number of access ports to the array.

For simple kernels such as Scramble, Descramble, SubCarrierMap, SubCarrierDemap, Modulation (WiBench), AntennaCombining, Windowing, MatchFilter, Interleave, and Demap (PHY-Bench), we can achieve a desirable performance with the primary directives. This is because the structures of these kernels are very simple. These kernels mostly contain single or multiple simple loops where they modify the data from the input array(s) and write the modified data to the output array(s). Therefore there is no need to optimize these kernels further. On the other hand, the Equalizer, Demodulation, RxRateMatch, and TxRateMatch kernels in WiBench and the CombinerWeights kernel in PHY-Bench are more complicated. They contain several sub-functions, and they are designed to be optimal for general-purpose processors. Therefore, although primary directives such as pipeline improve the performance of these kernels, we can improve them further without any significant effect on the FPGA resource utilization by using dataflow optimization. Dataflow optimization is a powerful directive that can take full advantage of parallelization and concurrency in the FPGA.

In dataflow optimization, the C/C++ code inside a function or loop must be partitioned into a set of sequential sub-functions. Then, HLS puts a memory channel between every two consecutive functions. There are buffers and FIFOs in each channel to store the data from the producer function and deliver them to the consumer function. Therefore, all functions can be executed in parallel, which improves the throughput and latency of the kernel. Although dataflow is an ideal solution, some behaviors in the C/C++ kernel need

to be resolved in order to use this directive. Some of the most important rules for dataflow optimization are: (1) no feedback between sub-functions; (2) no conditional execution between sub-functions; and (3) data should flow from one sub-function to the next, and the data cannot skip a sub-function. Another important point in dataflow optimization is that the code's throughput depends on the slowest function in the dataflow region. Therefore, the functions need to be partitioned carefully to ensure they have almost the same latency, to achieve the best performance. Therefore, we refactored the structure of the complex kernels of PHY-Bench and WiBench to harness the full potential of parallel execution in FPGA with dataflow optimization.

Finally, FFT and IFFT kernels from PHY-Bench and SCFDMADemodulation, SCFD-MAModulation, TransformDecoder, and TransformPrecoder kernels from the WiBench benchmark suite calculate discrete Fourier transforms (DFTs) to convert signals from the time domain to the frequency domain, and vice versa, using a fast Fourier transform (FFT) algorithm. Since FFT is widely used in different applications, Xilinx has already implemented this module efficiently. Therefore, although it is possible to use HLS to implement these kernels on the FPGA, the most efficient way is to use the FFT IP core provided by Xilinx.

Table 1 shows the latency and utilized resource of three implementations of each kernel for the PL part of the Ultrascale+ ZynqMP SoC (XCZU3EG-SBVA484). In the "No-Directive" implementation, we only made small changes to the C/C++ code to make the kernel synthesizable. The generated HDL modules had the highest execution time (latency), but they used fewer FPGA resources than the other implementations. In the "Primary-Directive" implementation, pipelining the loops in the code improved the latency of some kernels by up to 10 times. On the other hand, it increased the required FPGA resources by up to 2 times in some kernels. For instance, in the "Equalizer" kernel, the number of utilized DSPs increased from 18.89% to 34.44%. This is because, without primary directives, the synthesizer runs the loops sequentially and reuses the resources as much as possible. As mentioned earlier, some kernels achieved a desirable performance using the primary directive only. For more complicated kernels, Table 1 shows that compared to the "Primary-Directive" implementation, the "Dataflow" implementation, which used dataflow optimization, improved the performance of those kernels by up to 12 times. Table 1 shows that in the "Dataflow" implementation, the utilization of BRAM increased, because the synthesizer adds local buffers between sub-functions to increase parallelism. It is important to mention that our experiments show that different implementations of each kernel do not significantly affect the system's power consumption. We present the energy and power consumption of each kernel in Table 2.

The results show that the power consumption of the board does not change when the task is run on a single ARM Cortex-A53. This is because, when we run an application on the processor, one core is active during the execution and the power consumption is the same for all tasks (kernels). On the other hand, the number of FPGA resources that each kernel requires is different. This means the number of active cells in the FPGA during the execution of each task is different, which leads to dynamic power consumption.

**Table 1.** Latency and resource utilization of different kernels of PHY-Bench and WiBench benchmark suites on the PL part of Ultrascale+ Zynq with XCZU3EG-SBVA484.

| | Kernel | Implementation | Latency | | Resources (%) | | | |
|---|---|---|---|---|---|---|---|---|
| | | | Clk | Speedup [1] | BRAM | DSP | FF | LUT |
| **WiBench** | Equalizer (LAY = 2, ANT = 2, SYM = 14, MDFT = 75) | No-Directive | 619,276 | – | 0.00 | 18.89 | 17.83 | 38.09 |
| | | Primary-Directive | 607,351 | 1.02 | 0.00 | 34.44 | 22.54 | 44.86 |
| | | Dataflow | 95,371 | 6.49 | 2.78 | 47.78 | 25.82 | 42.07 |
| | Demodulation (LAY = 2, ANT = 2, SYM = 14, MDFT = 75) | No-Directive | 1,821,604 | – | 0.00 | 3.06 | 2.94 | 5.03 |
| | | Primary-Directive | 748,804 | 2.43 | 0.00 | 2.78 | 3.13 | 5.17 |
| | | Dataflow | 349,274 | 5.22 | 0.23 | 5.00 | 4.29 | 8.17 |
| | Modulation (LAY = 2, ANT = 2, SYM = 14, MDFT = 75) | No-Directive | 16,203 | – | 0.00 | 0.83 | 0.53 | 1.16 |
| | | Primary-Directive | 16,203 | 1.00 | 0.00 | 0.83 | 0.53 | 1.16 |
| | | Dataflow | 16,203 | 1.00 | 0.00 | 0.83 | 0.53 | 1.16 |

**Table 1.** *Cont.*

| | Kernel | Implementation | Latency | | Resources (%) | | | |
|---|---|---|---|---|---|---|---|---|
| | | | Clk | Speedup [1] | BRAM | DSP | FF | LUT |
| WiBench | Descramble (LAY = 2, ANT = 2, SYM = 14, MDFT = 75) | No-Directive | 64,803 | – | 0.00 | 1.67 | 0.52 | 0.69 |
| | | Primary-Directive | 7211 | 8.99 | 0.00 | 1.67 | 0.59 | 0.78 |
| | | Dataflow | 7211 | 8.99 | 0.00 | 1.67 | 0.59 | 0.78 |
| | Scramble (LAY = 2, ANT = 2, SYM = 14, MDFT = 75) | No-Directive | 14,403 | – | 0.00 | 0.83 | 0.33 | 0.45 |
| | | Primary-Directive | 7204 | 2.00 | 0.00 | 0.83 | 0.31 | 0.48 |
| | | Dataflow | 7204 | 2.00 | 0.00 | 0.83 | 0.31 | 0.48 |
| | RxRateMatch (LAY = 2, ANT = 2, SYM = 14, MDFT = 75) | No-Directive | 2486097 | – | 27.31 | 2.78 | 8.96 | 17.08 |
| | | Primary-Directive | 2,395,848 | 1.04 | 27.31 | 9.72 | 12.89 | 22.87 |
| | | Dataflow | 197,861 | 12.56 | 31.48 | 9.72 | 12.99 | 23.44 |
| | TxRateMatch (LAY = 2, ANT = 2, SYM = 14, MDFT = 75) | No-Directive | 1,923,250 | – | 27.31 | 4.44 | 9.29 | 17.77 |
| | | Primary-Directive | 1,907,485 | 1.01 | 27.31 | 10.00 | 13.08 | 23.25 |
| | | Dataflow | 158,678 | 12.12 | 31.25 | 15.28 | 20.60 | 35.35 |
| | SubCarrierDemap (LAY = 2, ANT = 2, SYM = 14, MDFT = 75) | No-Directive | 4992 | – | 0.46 | 0.56 | 0.87 | 1.93 |
| | | Primary-Directive | 2621 | 1.90 | 0.46 | 2.22 | 1.11 | 2.46 |
| | | Dataflow | 2621 | 1.90 | 0.46 | 2.22 | 1.11 | 2.46 |
| | SubCarrierMap (LAY = 2, ANT = 2, SYM = 14, MDFT = 75) | No-Directive | 6529 | – | 0.46 | 0.56 | 0.72 | 1.76 |
| | | Primary-Directive | 2355 | 2.77 | 0.46 | 2.50 | 0.99 | 2.11 |
| | | Dataflow | 2355 | 2.77 | 0.46 | 2.50 | 0.99 | 2.11 |
| PHY-Bench | AntennaCombining (LAY = 4, ANT = 4, SC = 1200) | No-Directive | 9601 | – | 0.00 | 4.44 | 0.47 | 0.63 |
| | | Primary-Directive | 4805 | 2.00 | 0.00 | 4.44 | 0.50 | 0.65 |
| | | Dataflow | 4805 | 2.00 | 0.00 | 4.44 | 0.50 | 0.65 |
| | Windowing (LAY = 1, ANT = 1, SC = 1200) | No-Directive | 2403 | – | 0.00 | 0.56 | 0.40 | 0.94 |
| | | Primary-Directive | 1208 | 1.99 | 0.00 | 0.56 | 0.44 | 1.01 |
| | | Dataflow | 1208 | 1.99 | 0.00 | 0.56 | 0.44 | 1.01 |
| | MatchFilter (LAY = 1, ANT = 1, SC = 1200) | No-Directive | 6001 | – | 0.00 | 1.67 | 0.33 | 0.45 |
| | | Primary-Directive | 1205 | 4.98 | 0.00 | 1.67 | 0.44 | 0.55 |
| | | Dataflow | 1205 | 4.98 | 0.00 | 1.67 | 0.44 | 0.55 |
| | CombinerWeights (LAY = 4, ANT = 4, SC = 100) | No-Directive | 149,601 | – | 0.46 | 8.89 | 7.09 | 15.43 |
| | | Primary-Directive | 123,501 | 1.21 | 0.69 | 15.00 | 8.91 | 22.03 |
| | | Dataflow | 28,964 | 5.17 | 3.80 | 18.83 | 13.13 | 24.96 |
| | Demap (LAY = 4, SYM = 6, SC = 50, MOD = 64QAM) | No-Directive | 134,401 | – | 0.00 | 3.33 | 1.74 | 4.82 |
| | | Primary-Directive | 28,809 | 4.67 | 0.00 | 3.33 | 2.57 | 9.13 |
| | | Dataflow | 28,809 | 4.67 | 0.00 | 3.33 | 2.57 | 9.13 |
| | Interleave (LAY = 1, SYM = 1, SC = 1200) | No-Directive | 4064 | – | 0.00 | 1.39 | 0.64 | 1.32 |
| | | Primary-Directive | 1267 | 3.21 | 0.00 | 1.39 | 2.16 | 8.03 |
| | | Dataflow | 1267 | 3.21 | 0.00 | 1.39 | 2.16 | 8.03 |

[1] Speedup with respect to No-Directive.

**Table 2.** The energy, power consumption, and execution time of different kernels on XCZU3EG-SBVA484 FPGA with 250 MHz clock frequency and ARM Cortex-A53 processor (only one core) with 1.5 GHz frequency.

| | Kernel | Latency (µs) | | HW Speedup | Power (mW) | | Energy (mJ) | |
|---|---|---|---|---|---|---|---|---|
| | | HW | SW | | HW | SW | HW | SW |
| WiBench | Equalizer | 495.9 | 6210.8 | 12.5 | 405.0 | 968.7 | 201 | 6016 |
| | Demodulation | 1760.3 | 21,694.2 | 12.3 | 375.0 | 968.7 | 660 | 21,015 |
| | Modulation | 111.5 | 1933.8 | 17.3 | 315.2 | 968.7 | 35 | 1873 |
| | Descramble | 41.0 | 764.1 | 18.7 | 315.2 | 968.7 | 13 | 740 |
| | Scramble | 54.5 | 771.2 | 14.2 | 315.2 | 968.7 | 17 | 747 |
| | RxRateMatch | 815.2 | 2702.6 | 3.3 | 405.0 | 968.7 | 330 | 2618 |
| | TxRateMatch | 641.1 | 2662.0 | 4.2 | 405.0 | 968.7 | 260 | 2579 |
| | Turbo Encoder | 385.6 | 1204.0 | 3.1 | 406.2 | 968.7 | 157 | 1166 |
| | SubCarrierDemap | 20.1 | 382.6 | 19.0 | 315.2 | 968.7 | 6 | 371 |
| | SubCarrierMap | 21.9 | 407.2 | 18.6 | 315.2 | 968.7 | 7 | 394 |

**Table 2.** *Cont.*

| | Kernel | Latency (μs) | | HW Speedup | Power (mW) | | Energy (mJ) | |
|---|---|---|---|---|---|---|---|---|
| | | **HW** | **SW** | | **HW** | **SW** | **HW** | **SW** |
| **PHY-Bench** | AntennaCombining | 19.8 | 339.1 | 17.1 | 405.0 | 968.7 | 8 | 328 |
| | Windowing | 5.4 | 40.5 | 7.4 | 315.2 | 968.7 | 2 | 39 |
| | MatchFilter | 5.5 | 83.3 | 15.1 | 315.2 | 968.7 | 2 | 81 |
| | CombinerWeights | 111.9 | 461.2 | 4.1 | 405.0 | 968.7 | 45 | 447 |
| | Demap | 115.8 | 2247.4 | 19.4 | 360.5 | 968.7 | 42 | 2177 |
| | Interleave | 5.5 | 53.2 | 9.7 | 315.2 | 968.7 | 2 | 52 |

## 5. Execution Time and Power Comparison for Hardware and Software on a Real Platform

This section describes how these kernels were implemented on a real platform to measure the latency and the power consumption. Figure 1 shows an overview of the ZynqMP SoC. The system includes a Zynq processing system, a direct memory access (DMA) [22], a module called "Kernel_Wrapper", and a couple of interconnects. We hid the clock and reset the signals in the figure for the sake of clarity. The "Kernel_Wrapper" contains an LTE kernel, multiple memories to store input and output data, and three AXI interfaces. In each kernel, there are two types of ports. The first type is responsible for receiving the input scaler data and the control signals from processor and sending back the output scaler data and status signals. The second type is responsible for input and output arrays in the kernel. These ports read and write data to local memories or FIFOs in the FPGA. The "Kernel_Wrapper" is the partially reconfigurable module of the design; therefore, it must have the same interface for all kernels. To this end, wrappers for all kernels have one AXI-Lite interface and two AXI-Stream interfaces. The processor configures the DMA and the kernel by reading the output scaler data and status signals and writing input scaler data and control signals using the AXI-Lite interface (brown wires in Figure 1). The "Kernel_Wrapper" has an AXI-Stream slave port that obtains the processor's data through the DMA and writes them to the input memories (blue wires in Figure 1). Finally, the "Kernel_Wrapper" has an AXI-Stream master port that sends back the results of the kernel, which are stored in the output memories of the processor through the DMA (red wires in Figure 1). In Figure 1, we only considered one PR region for running kernels. However, it is possible to increase the number of PR regions to improve the performance. For this purpose, for each additional slot, a DMA and a "Kernel_Wrapper" module must be added and connected to the processor using AXI interconnects.

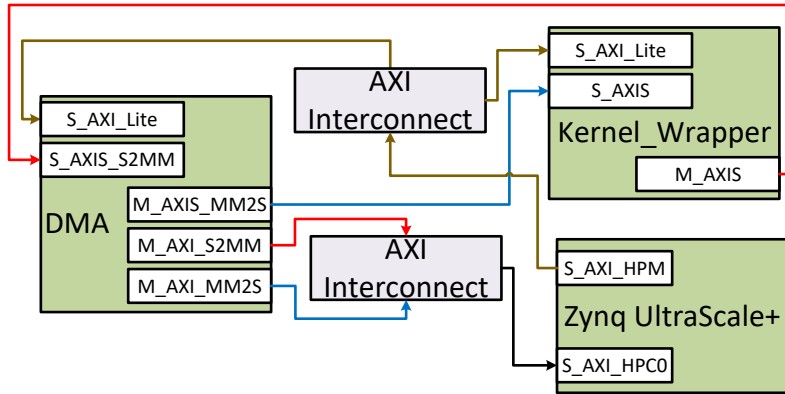

**Figure 1.** An overview of the system, including Zynq processor, DMA, Kernel_Wrapper module, and two interconnects.

Figure 2 shows the structure of the "Kernel_Wrapper" module. The data port of the AXI-Stream interface is connected to all the input memories. The processor first sets the chip enable (CE) pin of one of the input memories through the AXI-Lite interface and then starts the DMA engine to fill out the initial data. It repeats this procedure for all input memories. Then, the processor starts the kernel and checks for the done signal. When the LTE kernel sets the done signal, this means that the results are in the output memories. Then, the processor configures the selection bit of the multiplexer through AXI-Lite and reads the stored data in the output memory with the help of the DMA.

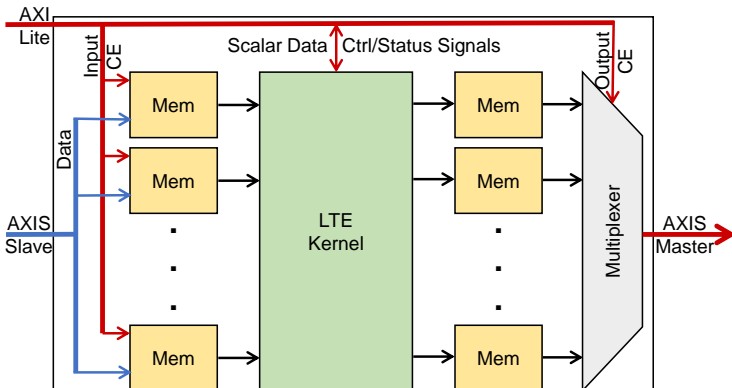

**Figure 2.** The structure of the Kernel_Wrapper module.

We executed each kernel on both the ARM processor (software) and FPGA (hardware), and we compared each kernel's execution time and power consumption. To this end, we exploited the Ultra96-V2 board by Avnet for running different kernels on the XCZU3EG-SBVA484 FPGA or the ARM Cortex-A53 processor and monitoring their realtime power consumption. Ultra96-V2 is an ARM-based, Xilinx Zynq UltraScale+ MPSoC development board with two power management units called "IRPS5401". These units are accessible through an IIC bus called "PMBus", and we were able to read the FPGA and ARM processors' voltage, current, power, and temperature separately using PMBus during the execution. The ARM Cortex-A53 works with a 1.5 GHz clock frequency, and the FPGA frequency is 250 MHz for all kernels.

The results in Table 2 show the effectiveness of the FPGA compared to ARM processors. It is important to mention that both ARM Cortex-A53 and XCZU3EG-SBVA484 FPGA are integrated on the same chip. We only used one core of the ARM Cortex-A53 in the PS. The power consumption of ARM Cortex-A53 was the same for all kernels and was 2.4 to 3 times higher than that of the FPGA. The main reason is that the frequency of the FPGA is much lower than that of the processor, and the kernels only occupy a small portion of the FPGA, while the rest of the FPGA is idle. Table 2 also shows that the execution time of each kernel on the FPGA was up to 19.4 times lower than on the ARM Cortex-A53 processor.

## 6. Partial Reconfiguration Effect

The partial reconfiguration feature delivers an effective solution for a more flexible HW/SW system with higher performance. In other words, to design a more efficient system, we needed partial reconfiguration to dynamically change the context of the FPGA during the run time. This is because the resources of the FPGA are limited and we could not statically implement all tasks. Therefore, with the help of partial reconfiguration, as for the PS, we could easily change the PL context and run more tasks on the PL. Hence, in the following subsections we will demonstrate the effects of PR with some experiments.

### 6.1. Effect of Partial Reconfiguration on Power Consumption: A Case Study on FFT

In this section, we present an experiment to show the effect of the module's size on the power consumption of the system and how it can be reduced with the help of partial reconfiguration. To this end, we considered two designs. In the first design, we instantiated

one FFT module on the partial reconfigurable region in the FPGA. We considered different scenarios, with a number of data samples in each frame of the FFT, called the transform length (TL), from 8 up to 4096. There was also another scenario where the system did not need to compute the FFT. Therefore, we considered an empty module without an FFT core but with the same interface. In the second design, we did the same, but instead of one FFT the PR region consisted of ten FFTs, all with the same transform length but different inputs. These scenarios are commonly used in LTE applications. For instance, in the uplink receiver application of PHY-Bench (Section 6.2), FFT's transform length must be equal to or bigger than the number of sub-carriers. Furthermore, the number of layers, antennas, and symbols affects the number of FFT modules that are needed.

We partially reconfigured the PL to execute FFTs with various transform lengths. The latency and power consumption values are presented in Table 3. The latency of both designs was the same because, in the second scenario, all the FFT modules were running in parallel. Table 3 shows that, for the design with a single FFT module, we could reduce the power consumption by up to 21% by using a suitable FFT (TL = 8) kernel instead of a large FFT (TL = 8192). For the design with ten FFT modules, the power could be reduced by up to 45%. Therefore, considering the input frame parameter, we could reconfigure a suitable FFT on the FPGA instead of using a large FFT to support all inputs. Additionally, Table 3 shows that the module with no FFT (idle case) still consumes a noticeable amount of power, due to the static part of the design. Therefore, if the size of the dynamic part of the design, which is changed during the partial reconfiguration, becomes much bigger than the size of the static part, as it does in the second design with ten FFTs, then the PR has a more dominant effect on the power consumption.

**Table 3.** The power consumption and latency of one-FFT and ten-FFT modules with various transform lengths.

| Transform Length | Latency (Clk) | Power Consumption of One FFT (mW) | Power Consumption of Ten FFTs All with the Same Transform Length (mW) |
|---|---|---|---|
| Idle | 0 | 312.50 | 312.50 |
| 8 | 94 | 343.75 | 343.75 |
| 16 | 146 | 343.75 | 343.75 |
| 32 | 242 | 343.75 | 375.00 |
| 64 | 434 | 359.25 | 375.00 |
| 128 | 834 | 359.25 | 406.25 |
| 256 | 1682 | 375.00 | 406.25 |
| 512 | 3490 | 375.00 | 437.50 |
| 1024 | 7346 | 375.00 | 437.50 |
| 2048 | 15,554 | 375.00 | 437.50 |
| 4086 | 32,978 | 406.25 | 500.00 |
| 8192 | 69,858 | 437.50 | 625.00 |

*6.2. Effect of Partial Reconfiguration on Time and Area: A Real-World Application Example*

In this section, we show the potential of partial reconfiguration to improve the execution time and area efficiency of the system with an example. Since exploring different scenarios on the board was time-consuming, we developed a Python script to find the best schedule that maps the tasks on HW and SW using the data extracted from the board. The Python script explores all the possible solutions and reports the best one. Figure 3 shows the SDF graph [23] of the LTE uplink receiver application from the PHY benchmark [9] for one-user equipment. The computation and latency of each kernel depend on various parameters such as the total number of layers (LAY), antennas (ANT), sub-carriers (SC), and symbols (SYM), and the modulation scheme (MOD). In Figure 3, the rectangles show how many times each actor needs to be fired to complete one iteration of the application. The figure also shows the parallelization factor (PF) per kernel [15]. For instance, if we

consider LAY = 2, ANT = 2, and SC = 1200, then ideally, we can parallelize the "Matched-Filter" kernel by a factor of $2 \times 2 \times 1200 = 4800$. Furthermore, the numbers on the arrows represent the data each actor consumes or produces in each run.

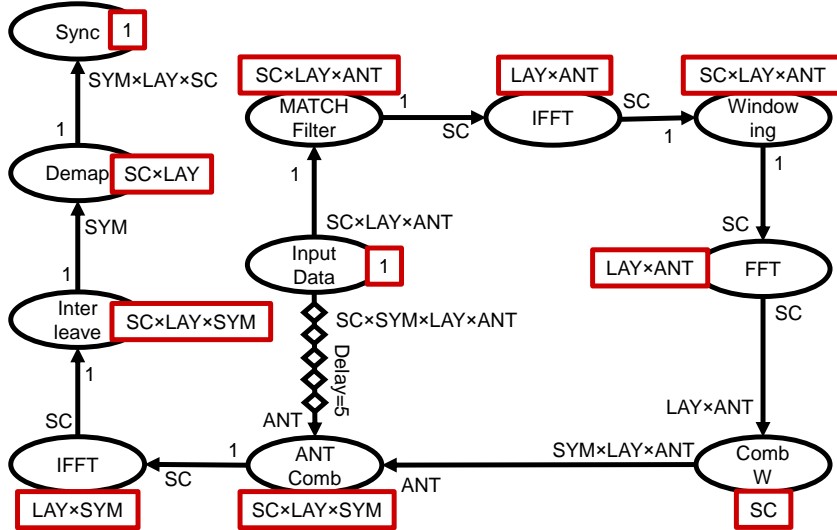

**Figure 3.** The SDF graph of an uplink stream application in PHY-Bench.

In uplink applications, the system needs to completely execute the graph shown in Figure 3 in 1 ms for each subframe. This experiment assumed that the system had 2 antennas, 2 layers, 6 data symbols, and up to 1200 subcarriers. In addition, the system used a 64QAM modulation scheme. The FFT and IFFT kernels are the bottleneck of this application. The FFT and IFFT nodes cannot start their execution before receiving all the sub-carriers (1200 in this example) of a layer and an antenna. In addition, it is impossible to break the operation into a smaller number of sub-carriers without affecting the functionality. The parallelization improves the system's performance; however, it increases the system's overhead, considering the additional logic required for scattering the input data and gathering the output data. Therefore, to provide an efficient design, we adjusted the input size of the other kernels to balance the execution time of all kernels. The results are presented in Tables 4 and 5. The second, third, and fourth columns of Table 4 show the execution time of each kernel, the number of times that kernel had to be executed, and the FPGA resources (maximum of LUT, FF, or DSP) used in each instance of the kernel, respectively.

**Table 4.** The result of running uplink application of PHY-Bench on XCZU3EG-SBVA484 FPGA for fully sequential and fully parallel strategies when LAY = 2, ANT = 2, SC = 1200, SYM = 6, and MOD = 64QAM.

| Kernel | Time (μs) | Runs (#) | Res (%) | Fully Sequential | | Fully Parallel | |
|---|---|---|---|---|---|---|---|
| | | | | Time (μs) | Res (%) | Time (μs) | Res (%) |
| MatchFilter | 19.2 | 1.0 | 2.0 | 19.2 | 2.0 | 19.2 | 2.0 |
| IFFT | 62.2 | 4.0 | 1.5 | 248.9 | 1.5 | 62.2 | 6.0 |
| Windowing | 19.2 | 1.0 | 1.0 | 19.2 | 1.0 | 19.2 | 1.0 |
| FFT | 62.2 | 4.0 | 1.5 | 248.9 | 1.5 | 62.2 | 6.0 |
| CombW | 44.2 | 20.0 | 8.0 | 884.8 | 8.0 | 44.2 | 160.0 |
| AntComb | 20.5 | 1.0 | 5.0 | 20.5 | 5.0 | 20.5 | 5.0 |
| IFFT | 62.2 | 12.0 | 1.5 | 746.6 | 1.5 | 62.2 | 18.0 |
| Interleave | 20.4 | 6.0 | 5.0 | 122.4 | 5.0 | 20.4 | 30.0 |
| Demap | 58.2 | 12.0 | 6.0 | 698.8 | 6.0 | 58.2 | 72.0 |
| Total Time (ms) | - | - | - | 3.0 | - | 0.4 | - |
| Total Res (%) | - | - | - | - | 31.5 | - | 300.0 |

**Table 5.** The results of running an uplink application of PHY-Bench on XCZU3EG-SBVA484 FPGA for partially parallel and partial reconfiguration strategies when LAY = 2, ANT = 2, SC = 1200, SYM = 6, and MOD = 64QAM.

| Kernel | Partially Parallel | | | Partial Reconfiguration | | |
|---|---|---|---|---|---|---|
| | Time (µs) | Res (%) | PF | Time (µs) | Res (%) | PF |
| MatchFilter | 19.2 | 2.0 | 1.0 | 19.2 | 2.0 | 1.0 |
| IFFT | 62.2 | 6.0 | 4.0 | 62.2 | 6.0 | 4.0 |
| Windowing | 19.2 | 1.0 | 1.0 | 19.2 | 1.0 | 1.0 |
| FFT | 62.2 | 6.0 | 4.0 | 62.2 | 6.0 | 4.0 |
| CombW | 221.2 | 32.0 | 4.0 | 177.0 | 40.0 | 5.0 |
| AntComb | 20.5 | 5.0 | 1.0 | 20.5 | 5.0 | 1.0 |
| IFFT | 186.6 | 6.0 | 4.0 | 62.2 | 18.0 | 12.0 |
| Interleave | 122.4 | 5.0 | 1.0 | 20.4 | 30.0 | 6.0 |
| Demap | 232.9 | 18.0 | 3.0 | 116.5 | 36.0 | 6.0 |
| Total Time (ms) | 0.9 | - | - | 0.6 | - | - |
| Total Res (%) | - | 81.0 | - | - | 80.0 | - |

The first approach was to instantiate only one instance for each kernel and run each kernel sequentially to complete the application. This approach only occupies 32% of the XCZU3EG-SBVA484 FPGA resources. However, it takes 3 ms to process a single sub-frame, which is not desirable. The second approach was to instantiate each kernel as many times as necessary and run all kernel instances in parallel. In this case, the execution time was 368.4 µs, which is less than 1 ms, satisfying the timing requirement. However, in this approach, we needed an FPGA with resources at least three times higher than XCZU3EG-SBVA484. The third approach (Table 5) was to unroll the kernel execution partially. The fourth column of Table 5 shows the parallelization factor for each kernel. For example, there were four CombW kernel instances and they were sequentially executed five times. This strategy achieved a 0.9 ms execution time with 81% utilization of the FPGA. Although the third approach satisfied both timing and area requirements, it was not scalable. For example, if we increased the number of antennas and layers from two to four, we could satisfy neither the timing nor the area requirements. The fourth approach was to use partial reconfiguration to improve the scalability of the third approach. To this end, we needed to set two partially reconfigurable regions (PRRs). Then, the system partially reconfigured PRR1 with the first kernel, which was MatchFilter. While the first kernel was running, the system partially reconfigured PRR2 with the second kernel, which was FFT. When the first kernel was executed, the second kernel in PRR2 started the execution, and the system partially reconfigured the third kernel in PRR1. Assuming that the PR time is less than the execution time of each kernel, we can hide the timing overhead for partial reconfiguration. This is a fair assumption for many applications in view of the speed of PCAP in recent Zynq ultra-scale FPGAs, which is approximately 450 MB/s. Furthermore, we can instantiate more instances of each kernel to further improve the timing of the application. Considering two PRRs, the system must fit the largest kernel (here this is CombW) into each PRR. Therefore, in this strategy, we achieved a 0.6 ms execution time with 80% utilization of the FPGA.

## 7. Conclusions

In this paper, we implemented well-known and useful LTE kernels on the FPGA using Vivado HLS. The use of HLS for generating these kernels makes it possible for users to modify, enhance, or test the kernels without interfering with the HDL code. We also refactored the structure of more complex kernels to apply dataflow optimization and improve the parallelism and performance of more complex kernels. We implemented all kernels on an Avnet Ultra96 board, and the results showed that executing the kernels on FPGA achieved an increase in speed of up to 19.4 times compared to running them on

ARM processors. Finally, we observed the effect of partial reconfiguration, and the results showed up to 45% power reduction. In addition, using PR, we could improve resource utilization, and the results showed that we could improve the execution time for processing a sub-frame by 33% compared to an FPGA-based approach without PR.

**Author Contributions:** Conceptualization, A.H. and A.K.; methodology, A.H. and A.K.; investigation, A.H. and A.K.; resources, A.H.; writing—original draft preparation, A.H.; writing—review and editing, A.K. and A.H.; supervision, A.K.; project administration, A.K.; funding acquisition, A.K. All authors have read and agreed to the published version of the manuscript.

**Funding:** This research received funding from the Truchard International Fund.

**Data Availability Statement:** The source code is publicly available at https://cfaed.tu-dresden.de/pd-downloads (accessed on 21 February 2022).

**Conflicts of Interest:** The authors declare no conflict of interest.

## Abbreviations

The following abbreviations are used in this manuscript:

| | |
|---|---|
| ANT | Antennas |
| ASIC | Application-Specific Integrated Circuit |
| AXI | Advanced eXtensible Interface |
| BRAM | Block Random Access Memory |
| CE | Chip Enable |
| DFT | Discrete Fourier Transform |
| DFX | Dynamic Function Exchange |
| DMA | Direct Memory Access |
| FF | Flipflop |
| FFT | Fast Fourier Transform |
| FIFO | First In First Out |
| FPGA | Field Programmable Gate Array |
| HDL | Hardware Description Language |
| HLS | High-Level Synthesis |
| IFFT | Inverse Fast Fourier Transform |
| IoT | Internet of Things |
| ILP | Integer Linear Programming |
| LAY | Layers |
| LTE | Long-Term Evolution |
| MOD | Modulation Scheme |
| MPSoC | Multi-Processor System on Chip |
| NoC | Network on Chip |
| PR | Partial Reconfiguration |
| PRR | Partially Reconfigurable Regions |
| RTL | Register Transfer Level |
| SC | Sub-Carriers |
| SDF | Synchronous Data Flow |
| SYM | Symbols |
| TL | Transform Length |
| WCDMA | Wideband Code Division Multiple Access |

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
