# Peer review of "A Partial-Reconfiguration-Enabled HW/SW Co-Design Benchmark for LTE Applications"

_electronics, doi:10.3390/electronics11070978_

Round 1

Reviewer 1 Report

Dear authors,

I reviewed your paper with interest.
There are some problems and questions about this paper.
Please answer the following questions and comments.

sincerely yours

-------------------------------------------------

1) I feel that the title of this paper is inappropriate.
From a hardware perspective, HW/SW co-design is possible because the FPGA used in this paper has both programmable logic (PL) and processor cores (PS). Also, from a software point of view, co-design is possible because, in addition to the hardware description language, high-level logic synthesis can be utilized, which makes it possible to use the programming languages such as C and C++.
On the other hand, I think that the partial reconfiguration of FPGA is not directly related to HW/SW co-design.

2) I feel that you are doing your best in your research, but the paper itself is not easy to understand. The main reason for this is that the results of the experiment are not described after explanation of the equipment and method used in  experiments, but the results are described suddenly.

3) How are the results in Chapter 3 obtained through experiments? There is no explanation of the equipment (software?) Used for the experiment and the procedure of the experiment. Please describe them before explaining the results. Please explain the three directives there as well.

4) How was the result of the PR in Chapter 5 obtained by experiment? After all, there is no explanation about the method and procedure of the experiment. Please describe them clearly.

5) The result of the PR is written in Table 5. But did you not measure the time required for PR of each kernels? If you measured, please write the results as well.

6) Considering the order of the chapters in the text, Table 2 should be presented before Figure 1.

7) The characters written in Figure 1 are too small to read. I think it is an auto-generated diagram by Vivado, but it should be rewritten for clarity.

8) In Figure 1, it is not explained that the central square is Zynq processing system.

9) There is no explanation of what the Zynq processing system is.

10) There is no explanation about AXI interface. It also doesn't explain the difference between AXI stream and AXI Lite.

11) It is not explained that the model number of a FPGA on the Ultra V2 board is XCZU3EG-SBVA484.

13) The FPGA model number should be written in uppercase, but in line 154 it is written in lowercase. (In lines 237 and 278, they are written in capital letters)

14) It is not explained that the ARM Cortex-A53 described in lines 71 and 210 is the processor core built in the FPGA.

15) In this paper, the term FPGA is used in the opposite sense to the processor core. However, the FPGA used in this study includes the PL and the processor core. Then PL should be used as the opposite term to the processor core. 

16) When each kernels are operated by software, the number of operated processor core is not written.

17) There is no explanation for PCAP.
-------------------------------------------------

Author Response

The authors would like to thank the reviewers for their valuable comments which helped us to improve the paper. For the sake of clarity, the reviewers’ questions/comments are highlighted in red and our responses in purple. Please note that all the references to Sections, Equations, Figures, etc. are based on their numbers in the revised version unless explicitly stated otherwise. Also, in the revised version of our paper, we have highlighted the changes using a blue-colored font.

Reviewer 2 Report

The paper presents an adaptive computing application for telecommunication. In the last thirty years the most exciting research area was the reconfigurable computing. The challenge to change an algorithm to another during run time named previously partial reconfiguration (which today is better called dynamic function exchange DFX or adaptive computing) evolved with the FPGA technology. Today the question is not how to reconfigure but what to reconfigure. The authors found a good field of the DFX application in telecommunications, which is also a very fast evolving technology.

The paper clearly stated what is the purpose of the research and what are the results. Background and related work are analyzed. No previous work cited from the university the work, however there are well known researchers in the field.

For the proposed implementations there are analyzed the two main problems performance and power consumption related to the consumed hardware resources.

From table 2 results that the processor is loaded uniformely, but the hardware power concuption sligthly differ. I would recommend to analyze this a little bit more and to conclude the difference, which should be the hardware resources.

The problem was deeply analyzed and presented. I would recommend after minor review to publication.

Author Response

(The authors gave the same response as above.)

Reviewer 3 Report

This manuscript presents the system architecture, parallelization exploration, and implementation of the kernels in PHY-Bench and WiBench. In addition, it discusses the impact of partial reconfiguration on the overall performance.

The paper is well-written and well-organized. However, still, a major issue remains. The Partial Reconfiguration is a major point as is reflected in the title. However, how the partial reconfiguration is achieved is not well introduced. Although it is not the major point of this manuscript, it is required to explain the effectiveness of the proposed solution.

Another issue is that, if the original input is well-designed HLS, then it is improper to state ' We developed an efficient HDL module for each kernel in ..'. These should be considered as the HLS based modules. An example reference could be found at https://dl.acm.org/doi/abs/10.1145/3289602.3293915

Author Response

(The authors gave the same response as above.)

Reviewer 4 Report

The paper explores efficiency aspects of Partial Reconfiguration (PR) / Dynamic Function Exchange (DFX), using PR to change a part of the FPGA functionality while other parts are working.

In this paper the authors develop an efficient HDL module for each kernel in PHY-Bench and WiBench benchmark suite using Vivado High-Level Synthesis tool. They had to refactor the C/C++ implementation 
to apply dataflow optimization.

An HDL wrapper is provided for each kernel with two AXI stream interfaces to receive input data and send output data through DMA.

Since the wrappers for all kernels have the same interface ports, it is possible to swap all kernels in the FPGA during run-time with the help of partial reconfiguration.

The authors also compared each kernel’s execution time and power consumption during execution on the ARM processor and on the FPGA. They used a Ultra96-V2 by Avnet (Arm-based, Xilinx Zynq UltraScale+ MPSoC development board) to run different kernels on FPGA and ARM Cortex-A53 processor.

In the paper they demonstrate the partial reconfiguration effect through two examples: (1) the effect of Partial Reconfiguration on power consumption with a case study on FFT; (2) the effect of Partial Reconfiguration on time and area with a real-world application example.

The text in the paper is well written and organized. The goal of this
work is clearly stated. Although rich in details, the paper is easy to follow.

The authors claim that the source code and the documentation will be public after acceptance at

https://cfaed.tu-dresden.de/pd-downloads,

so researchers can modify and regenerate all kernels with different sizes.

Author Response

(The authors gave the same response as above.)

Round 2

Reviewer 1 Report

Dear authors

I have read your replies for my questions and comments, and the revised paper.

You answered my questions and comments well, and your paper is well revised. Therefore, I think your paper is suitable to be published in the journal.

sincerely yours